# Expanding the Repertoire of the Plant-Infecting Ophioviruses through Metatranscriptomics Data

**DOI:** 10.3390/v15040840

**Published:** 2023-03-25

**Authors:** Humberto Debat, Maria Laura Garcia, Nicolas Bejerman

**Affiliations:** 1Instituto de Patología Vegetal, Centro de Investigaciones Agropecuarias, Instituto Nacional de Tecnología Agropecuaria (IPAVE-CIAP-INTA), Camino 60 Cuadras Km 5,5, Córdoba X5020ICA, Argentina; 2Unidad de Fitopatología y Modelización Agrícola, Consejo Nacional de Investigaciones Científicas y Técnicas, Camino 60 Cuadras Km 5,5, Córdoba X5020ICA, Argentina; 3Instituto de Biotecnología y Biología Molecular (IBBM-CONICET-UNLP), Facultad de Ciencias Exactas, Universidad Nacional de La Plata, Calle 50 y 115, La Plata 1900, Argentina

**Keywords:** plant viruses, ophiovirus, virus taxonomy, metatranscriptomics, virus discovery

## Abstract

Ophioviruses (genus *Ophiovirus*, family *Aspiviridae*) are plant-infecting viruses with non-enveloped, filamentous, naked nucleocapsid virions. Members of the genus *Ophiovirus* have a segmented single-stranded negative-sense RNA genome (ca. 11.3–12.5 kb), encompassing three or four linear segments. In total, these segments encode four to seven proteins in the sense and antisense orientation, both in the viral and complementary strands. The genus *Ophiovirus* includes seven species with viruses infecting both monocots and dicots, mostly trees, shrubs and some ornamentals. From a genomic perspective, as of today, there are complete genomes available for only four species. Here, by exploring large publicly available metatranscriptomics datasets, we report the identification and molecular characterization of 33 novel viruses with genetic and evolutionary cues of ophioviruses. Genetic distance and evolutionary insights suggest that all the detected viruses could correspond to members of novel species, which expand the current diversity of ophioviruses ca. 4.5-fold. The detected viruses increase the tentative host range of ophioviruses for the first time to mosses, liverwort and ferns. In addition, the viruses were linked to several *Asteraceae*, *Orchidaceae* and *Poaceae* crops/ornamental plants. Phylogenetic analyses showed a novel clade of mosses, liverworts and fern ophioviruses, characterized by long branches, suggesting that there is still plenty of unsampled hidden diversity within the genus. This study represents a significant expansion of the genomics of ophioviruses, opening the door to future works on the molecular and evolutionary peculiarity of this virus genus.

## 1. Introduction

A vast number of viruses are being discovered in this new metagenomic era, revealing a multifaceted and diverse evolutionary landscape of replicating entities and the complexities associated with their arduous classification [1]. Several strategies to lever this dynamically growing wide-ranging assemblage of viruses have led to an initial comprehensive proposal to generate a virus world megataxonomy [2]. Despite extensive and broad efforts to characterize the virus share of the biosphere, only an infinitesimal portion, which probably embodies less than one percent of the virosphere, appears to be characterized so far [3]. Consequently, our knowledge about the massive global virome, with its outstanding diversity and including every prospective host organism assessed so far, is scarce [4,5,6]. Data mining of publicly available transcriptome datasets derived from high-throughput sequencing (HTS) has become an efficient and inexpensive strategy to uncover the hidden diversity of the plant virosphere [5,7]. Data-driven virus discovery emerges in the context of a massive number of open datasets in the Sequence Read Archive (SRA) of the National Center for Biotechnology Information (NCBI). This wonderful reserve of sequences, which is growing at an exceptional rate, represents a substantial (but still limited and biased) portion of all the organisms that populate our world, and the NCBI-SRA database is an efficient and cost-effective resource to identify novel viruses [8]. From a virus taxonomy perspective, a consensus statement has defined that viruses that are known only from metagenomic data can, and should, be incorporated into the official classification scheme of the International Committee on Taxonomy of Viruses (ICTV) [9].

Ophioviruses (genus *Ophiovirus*, family *Aspiviridae*) are plant-infecting viruses with non-enveloped, filamentous, naked nucleocapsid virions. Members of the genus *Ophiovirus* have a segmented single-stranded negative and possible ambisense RNA genome, encompassing three or four linear segments (in total ca. 11.3–12.5 kb) [10]. These segments encode four to seven proteins in the sense and antisense orientation, both in the viral and complementary strands [10]. The genus *Ophiovirus* includes seven recognized species with viruses infecting both monocots and dicots, mostly trees, shrubs and some ornamentals, and four out of these seven species are reported to be transmitted via soil-borne fungus of the genus *Olpidium spp* [10]. From a genomic perspective, as of today, there are complete genomes available for only four of these seven member species. In the context of a systematic expansion of virus discovery supported by the extensive use of HTS, a plethora of novel viruses of many families from diverse plants has been described. Nevertheless, to our knowledge, the diversity of ophioviruses appears to have stagnated, with no new ophiovirus species recognized by the ICTV since 2015. Two recent works have described the complete genome of a novel proposed ophiovirus associated with carrot, carrot ophiovirus 1 (CaOV1) [11], and another found in pepper, pepper chlorosis-associated virus (PCaV) [12]. In addition, the segment that encodes the capsid protein (CP) of a putative novel ophiovirus was assembled from transcriptomic data of *Dactylorhiza hatagirea* [13].

This is the first study oriented to identify and characterize ophiovirus sequences that are hidden in publicly available metatranscriptomic data, which resulted in the identification and characterization of 33 novel tentative ophioviruses. Our findings significantly expand the status quo of the genomics of ophioviruses, opening the door to future works on the molecular and evolutionary peculiarities of this virus genus and the *Aspiviridae* family.

## 2. Material and Methods

### 2.1. Identification of Ophiovirus Sequences from Public Plant RNA-Seq Datasets

Two strategies were used to detect ophiovirus sequences. (1) Assembled and raw sequence data corresponding to the 1K study [14] were explored using tBlastn searches (E-value < 1e^−5^) for ophiovirus sequences using the NCBI-refseq proteins of ophioviruses in the 1KP:BLAST tool (https://db.cngb.org/onekp, accessed on 20 January 2023), and hits were curated with the raw SRA data retrieved from the NCBI BioProject PRJEB4922. (2) The Serratus database was analyzed, employing the serratus explorer tool [5] using as the query the predicted RNA-dependent RNA polymerase protein (RdRP) of ophioviruses available in the NCBI-refseq database. The SRA libraries that matched the query sequences (alignment identity > 45%; score > 10) were further explored in detail.

### 2.2. Sequence Assembly and Virus Identification

Virus discovery was implemented as described elsewhere [15,16]. In brief, the raw nucleotide sequence reads from each SRA experiment that matched the query sequences in both the 1k and Serratus platforms were downloaded from their associated NCBI BioProjects (Table 1). The datasets were pre-processed by trimming and filtering with the Trimmomatic v0.40 tool as implemented in http://www.usadellab.org/cms/?page=trimmomatic, accessed on 20 January 2023 with standard parameters except quality required, which was raised from 20 to 30 (initial ILLUMINACLIP step, sliding window trimming, average quality required = 30). The resulting reads were assembled de novo with rnaSPAdes using standard parameters on the Galaxy server (https://usegalaxy.org/, accessed on 20 January 2023). The transcripts obtained from the de novo transcriptome assembly were subjected to bulk local BLASTX searches (E-value < 1e^−5^) against ophiovirus refseq protein sequences available at https://www.ncbi.nlm.nih.gov/protein?term=txid88129[Organism], accessed on 20 January 2023. The resulting viral sequence hits of each dataset were explored in detail. Tentative virus-like contigs were curated (extended and/or confirmed) by iterative mapping of each SRA library’s filtered reads. This strategy is used to extract a subset of reads related to the query contig, use the retrieved reads from each mapping to extend the contig and then repeat the process iteratively using as query the extended sequence. The extended and polished transcripts were reassembled using the Geneious v8.1.9 (Biomatters Ltd., Boston, MA, USA) alignment tool with high sensitivity parameters. 

### 2.3. Bioinformatics Tools and Analyses

#### 2.3.1. Sequence Analyses

ORFs were predicted with ORFfinder (minimal ORF length 150 nt, genetic code 1, https://www.ncbi.nlm.nih.gov/orffinder/, accessed on 20 January 2023) and the functional domains and architecture of translated gene products were determined using InterPro (https://www.ebi.ac.uk/interpro/search/sequence-search, accessed on 20 January 2023) and the NCBI Conserved domain database-CDD v3.20 (https://www.ncbi.nlm.nih.gov/Structure/cdd/wrpsb.cgi, accessed on 20 January 2023) with e-value = 0.01. Furthermore, HHPred and HHBlits as implemented in https://toolkit.tuebingen.mpg.de/#/tools/, accessed on 20 January 2023 were used to complement the annotation of divergent predicted proteins with hidden Markov models. Transmembrane domains were predicted using the TMHMM version 2.0 tool (http://www.cbs.dtu.dk/services/TMHMM/, accessed on 20 January 2023). The predicted proteins were then subjected to NCBI-BLASTP searches against the non-redundant protein sequences (nr) database to filter out any virus-like sequences that did not show an ophiovirus protein as best hit.

#### 2.3.2. Pairwise Sequence Identity

Percentage amino acid (aa) sequence identities of the predicted CP protein of the ophioviruses identified in this study, as well as those available in the NCBI database, were calculated using SDTv1.2 [42] based on MAFFT 7.505 (https://mafft.cbrc.jp/alignment/software, accessed on 20 January 2023) alignments with standard parameters. Virus names, abbreviations and NCBI accession numbers of ophioviruses already reported are shown in Appendix A.

#### 2.3.3. Phylogenetic Analysis

Phylogenetic analysis based on the predicted CP protein or the polymerase protein of all available ophioviruses was carried out using MAFFT 7.505 with multiple aa sequence alignments using G-INS-i and E-INS-i as the best-fit model, respectively. The aligned aa sequences were used as input to generate phylogenetic trees through the maximum-likelihood method with the FastTree 2.1.11 tool available at http://www.microbesonline.org/fasttree/, accessed on 20 January 2023. Local support values were calculated with the Shimodaira–Hasegawa test (SH) and 1000 tree resamples. The capsid proteins of two selected cytorhabdoviruses (alfalfa dwarf virus YP_009177015 and lettuce necrotic yellows virus YP_425087) were used as the outgroup in the CP tree. The polymerase proteins of three related and unclassified aspivirus-like viruses (nees’ pellia aspi-like virus CAH2618860, Plasmopara viticola lesion ass. mycoophiovirus 1 QJX19787, grapevine-associated serpento-like virus 1 QXN75438) were used as the outgroup in the polymerase trees. To explore the potential phylogenetic co-divergence of ophioviruses with their associated host plants, plant host cladograms were generated in phyloT v.2 (https://phylot.biobyte.de/, accessed on 20 January 2023), based on NCBI hierarchical taxonomy. Host associations were based on connections manually inferred between viral and plant phylogram and cladograms.

## 3. Results 

### 3.1. Summary of Discovered Ophiovirus Genomic Sequences

In this study, through the identification, assembly and curation of raw NCBI-SRA reads of publicly available transcriptomic data, we identified genomic evidence of 33 novel ophioviruses. Full-length viral genome sequences were obtained for 12/33, and 5/33 of the putative viruses had all their RNA segments detected, while 16/33 had some missing, mostly derived from the technical difficulties of assembling segments that are at relatively low RNA levels during infection such as RNA 1 (Table 1, Appendix A). Importantly, 85% of the identified viruses included the detection of two or more RNA segments of the virus in the same sequencing library, which improved the level of confidence in the discovery. The detected viruses were associated with 33 different plant host species (Table 1). The majority of the host plants were herbaceous dicots, with 20 out of 33 identified as such. The remaining hosts were herbaceous monocots, liverworts, mosses and ferns (Table 1). The genomes of 15 out of 17 viruses with all RNA segments annotated had three segments, while two monocot-associated ophioviruses had four segments (Table 1, Figure 1). 

### 3.2. Structural and Functional Annotation of Ophiovirus Sequences

The RNA segments of the detected viruses were found to encode various proteins, including the polymerase, movement protein, and capsid protein. The RNA 1 encoded two proteins at 3′ of the vcRNA, a large 261–280 kDa protein including the core polymerase module with the typical conserved motifs “A–E” of the RdRP, with the expected SDD signature sequence in motif “C” (Mononeg_RNA_pol, pfam00946). Separated by an intergenic region, the other ORF at 5’ of the vcRNA, encoded a small protein with a size that ranged from 105 to 245 amino acids (aa) (Figure 1). Interestingly, this small protein was quite diverse in most of the viruses identified in this study, and no hits were found when BLASTP searches were conducted (Table 1). The vcRNA 2 encoded a putative movement protein (MP) ranging from 47 to 58 kDa, and all the predicted MP proteins presented the 30K core MP domain (30K_MP, pfam17644). In addition, a few detected viruses encoded a small 6–10 kDa protein in the vRNA 2 with no blast hits or conserved domains, supporting the possibility of the ambisense coding strategy suggested for MLBVV. The vcRNA 3 encoded the capsid protein [10,43], ranging from 48–57 kDa and presenting an ssRNA negative plant viral coat protein nucleocapsid domain (Nucleocap, pfam11128) and no additional ORFs (Figure 1). The RNA 4 encoded a protein with unknown function with a size that ranged between 322 and 360 aa, in some instances including an overlapped ORF encoding a 10–12 kDa protein of unknown function. Nuclear localization signals were also found in the polymerase, MP and CP encoded by the viruses identified in this study

### 3.3. Pairwise Identities of Ophiovirus Sequences and Species Demarcation Criteria

The pairwise aa sequence identities between the CP proteins of all reported ophioviruses, including those identified in this study, showed great diversity with an identity ranging from 14.2% to 98.9%, but importantly with a mean identity of only 32.1% (Appendix A). Using the molecular criterion for species demarcation threshold of 85% aa identity of the CP [10], all ophioviruses with complete CP coding regions assembled in this study with an identity below 85% were tentatively deemed to be members of new ophiovirus species (Appendix A), increasing the number of potential members of the genus more than 4.5-fold. We suggest potential latinized binomial virus species names to include the viruses described here as members of novel species within the genus *Ophiovirus* (Table 2). 

### 3.4. Phylogenetic Relationships between Ophioviruses and Hosts

Phylogenetic analyses based on the deduced CP protein aa sequences of the detected viruses revealed a complex evolutionary history, showing distinctive groups and associations (Figure 2). One cluster included a group of 11 viruses with affinities to BlMaV, six to CPsV and a novel basal group of two viruses detected in *Asteraceae*-plants (Figure 2). The other known clade of five ophioviruses was expanded with two grass viruses with affinities to LRNV, and the recently reported CaOV1 and PCaV were linked to the MLBVV/TMMMV group and the freesia sneak virus (FreSV) and ranunculus white mottle virus (RWMV) group, respectively. More distantly, three small groups of viruses were found including four new viruses of orchids, and the third most basal group with very large branches of a virus associated with a poacea and another one with the aquatic plant *Zostera japonica*. Furthermore, a novel divergent clade was found, mostly represented by viruses detected in basal plants such as mosses, liverworts and ferns (Figure 2). Additional phylogenetic analyses based on the deduced RdRP protein aa sequences showed a similar evolutionary history of the corresponding viruses to the one predicted with the CP protein (Appendix A), that is, shared local clustering of many viruses indicating co-divergence in both the CP and RdRP trees, consistent with a common phylogenetic trajectory (Appendix A). In addition, we generated a tanglegram to compare the virus phylogram and plant host cladogram to further explore potential virus–host relationships (Figure 3 and Appendix A). This analysis showed that viruses of some clades clearly co-diverged with their hosts, including an orchid-associated virus clade and a clade of fern, moss and liverwort viruses (Figure 3 and Appendix A).

## 4. Discussion

### 4.1. Discovery of Novel Ophioviruses Expands Their Diversity and Evolutionary History

Known ophioviruses are agronomically relevant, including viruses generating detrimental infections and disease in crops and ornamental plants. This status quo is grounded on a tradition of biased sampling oriented to virus discovery in symptomatic and economically important plants. In this scenario, ophiovirus presence is not expected in the sequencing libraries of non-symptomatic vegetables; thus, they are ideal candidates to be identified through the mining of publicly available metatranscriptomic data. However, in the context of massive efforts directed to virus discovery in plants, as of today, only the partial genome of just one novel tentative ophiovirus was discovered when publicly available transcriptome datasets were mined [13]. Therefore, to assess whether this apparently limited ophiovirus diversity was biological or technical, we directed our efforts to specifically address ophiovirus discovery. We extensively searched for these viruses in already available plant transcriptome datasets to expand the repertoire of plant-infecting ophiovirus. This in silico-driven search resulted in the identification of virus sequence evidence of 33 novel ophioviruses. We also detected three novel variants of members of two known ophiovirus species. This substantial number of newly discovered putative ophioviruses represents a 4.5-fold increase in the known ophioviruses, which undoubtedly shows the importance of data-driven virus discovery to expand our understanding of the genomic diversity and peculiarities of virus taxa, such as the ophiovirus.

### 4.2. Host Range and Genomic Organization of the Novel Ophioviruses

Most of the host plants in which the novel viruses of this study were identified are herbaceous dicots, which, overall, are the most common hosts of known ophioviruses. Ophioviruses were detected in liverworts, mosses and ferns for first time, thus expanding the host range of these viruses. Only two viruses with all RNA segments annotated had four segments, which is also a genomic organization of the ophioviruses Mirafiori lettuce big-vein virus (MLBVV), lettuce ring necrosis virus (LRNV) [10] and the recently reported carrot ophiovirus 1 [11]. Thus, the most frequent genomic organization found for ophioviruses consists of three RNA segments. 

### 4.3. Genomic Features of the Discovered Ophioviruses

Like all previously reported ophioviruses [10], the RNA1 encoded the polymerase and a small protein. The RNA 1 small protein of the citrus psorosis virus (CPsV), the 24K protein, has been described to localize at the nucleus, is involved in miRNA misprocessing in citrus [44] and is an RNA-silencing suppressor [45]. The RNA2 encoded the putative MP, which was characterized as a cell-to-cell MP for CPsV (54K protein) and MLBVV (55K protein) [46,47]. All the predicted MP proteins detected presented the 30K core MP domain including the signature aspartate involved in cell-to-cell movement [48]. In addition, in the vRNA2, a highly divergent small protein was found to be encoded by few of the identified viruses, which is consistent with the proposed ambisense nature of RNA2 postulated for MLBVV, which harbors a 10 kDa protein of unknown function at the same locus [49]. Further, the RNA3 encoded the CP [10,43], with its typical ssRNA negative nucleocapsid domain. The RNA 4, which we identified only in three monocot-associated viruses, encoded a protein with unknown function. MLBVV RNA 4 contains a second overlapping ORF with no initiation codon and is proposed to be expressed by a + 1 translational frameshift, encoding a 10.6 kDa protein [49]. We failed to detect a similar additional overlapped ORF in the identified viruses, but we tentatively annotated a small ORF encoding a 12 kDa protein that was separated by an intergenic region at 3´of the vcRNA 4 of Agrostis ophiovirus, which was conserved in the virus sequences of both plant hosts where these viruses were detected. Similarly to what was previously reported for ophioviruses [10], we identified nuclear localization signals in the polymerase, MP and CP encoded by the ophioviruses identified in this study. 

### 4.4. Sequence Diversity and Evolutionary Clues of Identified Ophioviruses

A great diversity was found within the pairwise aa sequence identities between the CP proteins of all reported ophioviruses, including those identified in this study. The overall low sequence identity determined suggests that there is likely a substantial amount of undiscovered ophioviruses that may inhabit this virus space, despite the numerous viruses identified in this study. The genetic distance assessment was complemented with phylogenetic insights to provide evolutionary clues of the identified viruses.

Previous studies placed the ophiovirus in two distinct clades, one including a closer relationship between MLBVV and tulip mild mottle mosaic virus (TMMMV) and a separate clade conformed by blueberry mosaic-associated virus (BlMaV) and CPsV. These two are placed more distantly to the other ophioviruses, suggesting that this might lead to the re-assignment of the existing species into two separate genera [10]. On the one hand, the long branches linking BlMaV and CPsV in previous analyses [10] undoubtedly constituted viral “dark matter”, as at least 19 new viruses expand the bounds of the viral sequence space between these two viruses, including a novel basal group of two viruses detected in *Asteraceae* plants. The other clade was expanded with two grass viruses with affinities. Three small groups of viruses were found with a distant evolutionary history, including a virus associated with the aquatic plant *Zostera japonica*. Interestingly, a few years ago, the first endogenous sequence of an ophiovirus was detected in the genome of the related eelgrass *Zostera marina* [50]. In the genome of this plant, a CP-like sequence was found, flanked by transposable elements, suggesting an ancient shared evolutionary history of eelgrass and ophioviruses, and the possibility that this group of plants might host contemporary ophioviruses, which is in line with the detected virus hosted by eelgrass in this work. Moreover, we found a novel divergent clade that consisted of viruses associated with basal plants such as mosses, liverworts and ferns, which represents the first association of ophioviruses with non-vascular plants and pteridophytes. The phylogenetic analyses based on the deduced RdRP protein aa sequences showed a similar evolutionary history of the corresponding viruses, supporting the results based on CP assessment. For instance, fern-, moss- and liverwort-associated ophioviruses clustered together both in CP- and RdRP-based trees, suggesting that they share a unique evolutionary history among ophioviruses. The tanglegram showed that the orchid-associated virus clade and the clade of fern, moss and liverwort viruses clearly co-diverged with their hosts, suggesting a shared host–virus evolution in these groups. Nevertheless, the tanglegram topology also showed that for many of the ophioviruses, there is no apparent concordant evolutionary history with their potential plant hosts. 

### 4.5. Ophiovirus Tentative Taxonomical Classification

The distinctive phylogenetic clustering and the significant divergence in terms of aa identity of the predicted proteins of several of the identified viruses raises questions about taxonomic classification. Currently, the family *Aspiviridae* includes a single recognized genus with seven member species, and following the molecular criterion for ophiovirus species demarcation of a CP amino acid sequence identity <85%, we suggest that all the identified viruses in this study could be members of novel species, which were named based on current guidelines [51]. Nevertheless, it has not escaped our notice that eventually, some of the groups of viruses reported here, if recognized, could be included in new genera within the *Aspiviridae* family, applying a genus demarcation criterion still not defined. The outstanding divergence we found in some identified viruses highlights the need for novel approaches to classify this emerging ophio-like virus diversity. For instance, a percentage CP identity threshold could also be defined as a genus demarcation criterion (e.g., <40–45%), which should be integrated with predictions based on phylogenetic insights. Moreover, the existence of unclassified aspi-like viruses reported with as yet unknown CP predicted proteins raises the possibility of using other genetic markers. One possibility to define subfamilies within *Aspiviridae* could be implemented by using an identity threshold of the RdRP as a molecular criterion (e.g., <30% identity), as is the case for several RNA virus families.

### 4.6. Potential Vectors and Transmission Modes

Members of four out of the seven ophiovirus species recognized so far are reported to be transmitted via soil-borne fungus of the genus *Olpidium* [10], while for CPsV, which is transmitted by vegetative propagation of the host, no natural vector had been identified [10]. Nevertheless, while we assessed thousands of sequencing libraries in the Serratus platform, we failed to robustly detect ophiovirus-like sequences in any fungal library. Interestingly, one of the ophioviruses identified in this study was discovered in a transcriptome dataset of bumblebees. Further inspection of the raw reads of this dataset retrieved a significant amount of plant reads, which, based on rRNA analysis, corresponded to the *Boraginacea* family. We tentatively linked this virus to this family of plants, and we cautiously speculate on the possibility that this ophiovirus could be pollen-associated and transported to other plants by bumblebees. In this line, a recent study characterized the pollen virome of wild plants, identifying plenty of pollen-associated viruses, but no ophioviruses [52]. Moreover, these authors found that the pollen virome is visually asymptomatic. This anecdotal observation and our difficulties in detecting ophiovirus-like sequences in fungal libraries could provide some grounds for the possibility that a share of ophioviruses could be vertically transmitted. Other lines of evidence could support this suggestion: i) host–virus co-divergence in some clades may implicate isolation and a lack of horizontal transmission and ii) an emerging characteristic persistent, chronic infections of several plant viruses that are vertically transmitted are latent/asymptomatic infections, a feature that could be shared by ophioviruses. Thus, further studies should be carried out to elucidate alternative transmission modes of ophioviruses beyond the fungally transmitted MLBVV, TMMMV, LRNV and FreSV [53,54]. 

### 4.7. Limitations of Sequence Discovery through Data Mining

There are many limitations in this study, for instance, the incapacity to return to the original biological material to repeat and check the assembled viral genome sequences is a noteworthy restriction of the data mining approach for virus discovery. Another restriction is derived from difficulties during the assembly of genome segments represented at relatively low viral RNA titters in sequencing libraries (e.g., RNA 1). This resulted in many detections where we failed to assemble complete or nearly complete genomes, or where the level of confidence on the consensus sequence is lower. The reader may find Appendix A useful to assess the robustness of each identified virus sequence based on several metrics. Similarly, contamination, low sequencing quality, spill over and other technical artefacts could result in false positive detections, chimeric assemblies or poor host assignment. New RNAseq datasets derived from the predicted plant hosts would definitely improve and complement our results. In addition, a lack of a directed strategy to address virus segment termini, such as RACE, results in difficulties in determining bona fide RNA virus ends, which have conserved functional and structural cues in ophioviruses [10]. Some aspects of our strategy for virus discovery can overcome several of these limitations, providing additional evidence on identification, for instance, the detection of the same putative virus in independent libraries from the same plant host, a robust depth coverage of virus reads, the detection of more than one RNA segment of the virus in the same library or the detection of strains of a virus in evolutionarily related plants. Nevertheless, associations and detections should be complemented by further studies. 

## 5. Conclusions

In summary, this study illustrates the significance of the analysis of NCBI-SRA public data as a valued tool to not only accelerate the discovery of novel viruses but also to increase our understanding of their evolution and to improve virus taxonomy. Using this approach, we looked for hidden ophio-like virus sequences to expand the repertoire of these viruses, expanding the potential existing members within the genus4.5-fold. Additionally, we fostered the most comprehensive phylogeny of ophioviruses to date and shed new light on the phylogenetic relationships and evolutionary landscape of this group of viruses. Future studies should focus not only on complementing our genomic predictions, but also on providing clues for the biology and ecology of these viruses such as associated symptomatology, transmission and putative vectors. 

## Figures and Tables

**Figure 1 viruses-15-00840-f001:**
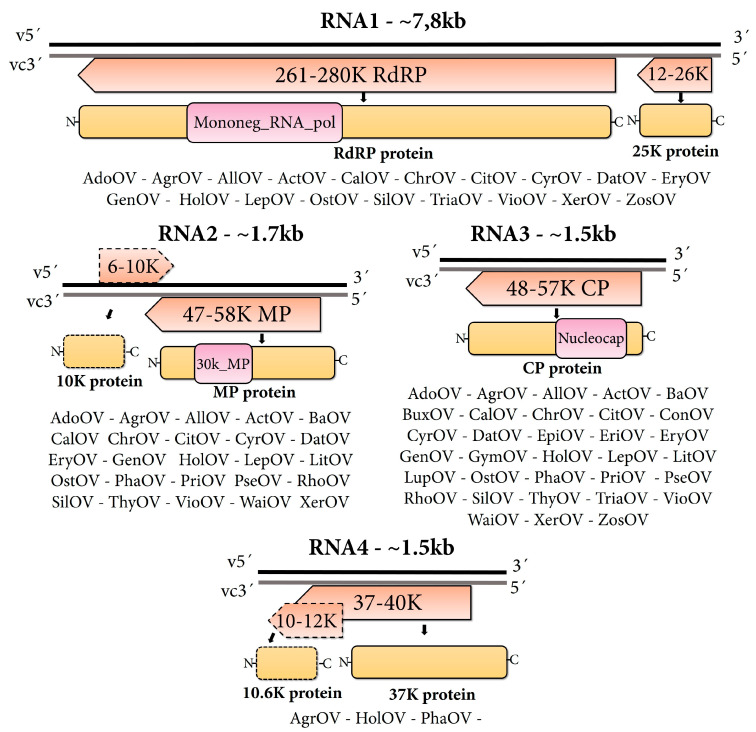
Genomic architecture of ophioviruses detected in this work. Genome graphs depicting organization and predicted gene products of each RNA segment. The predicted coding sequences are shown in orange arrowed rectangles. Gene products are depicted in curved yellow rectangles and their name is indicated below based on the general genome architecture. Dotted rectangles represent less common ORFs. Sizes in nucleotides and molecular weights in kilo Daltons of predicted proteins are indicated. Abbreviations: CP, capsid protein CDS; R, RNA-dependent RNA-polymerase CDS; MP, movement protein; v, virus RNA strand; vc, virus complementary RNA strand. Virus abbreviations are described in Table 1.

**Figure 2 viruses-15-00840-f002:**
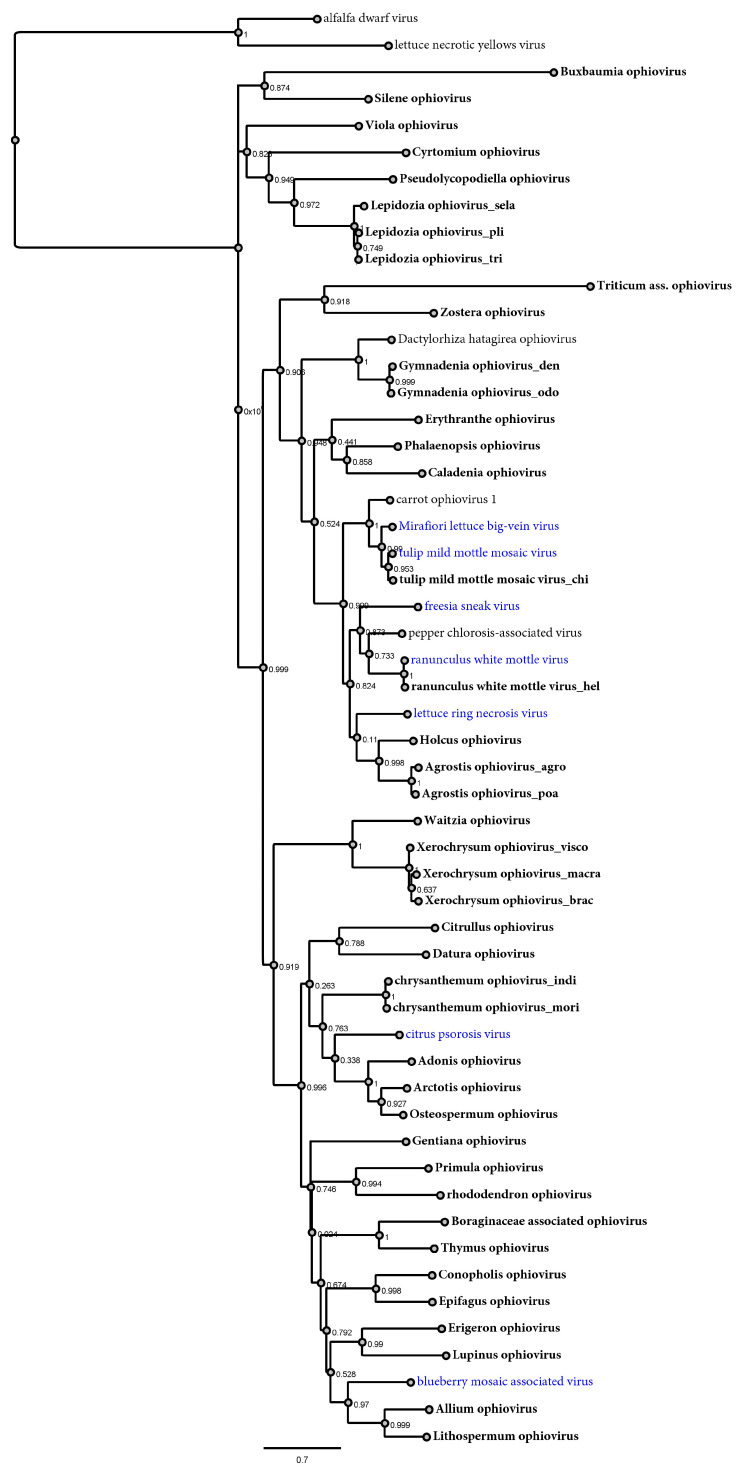
Maximum-likelihood phylogenetic tree based on the amino acid MAFFT sequence alignments of the CP protein of all the ophioviruses reported thus far and in this study. The scale bar indicates the number of substitutions per site. The node labels indicate FastTree support values. The CP proteins of two cytorhabdoviruses (alfalfa dwarf virus YP_009177015 and lettuce necrotic yellows virus YP_425087) were used as outgroups. Viruses corresponding to members of ICTV-recognized species are depicted in blue.

**Figure 3 viruses-15-00840-f003:**
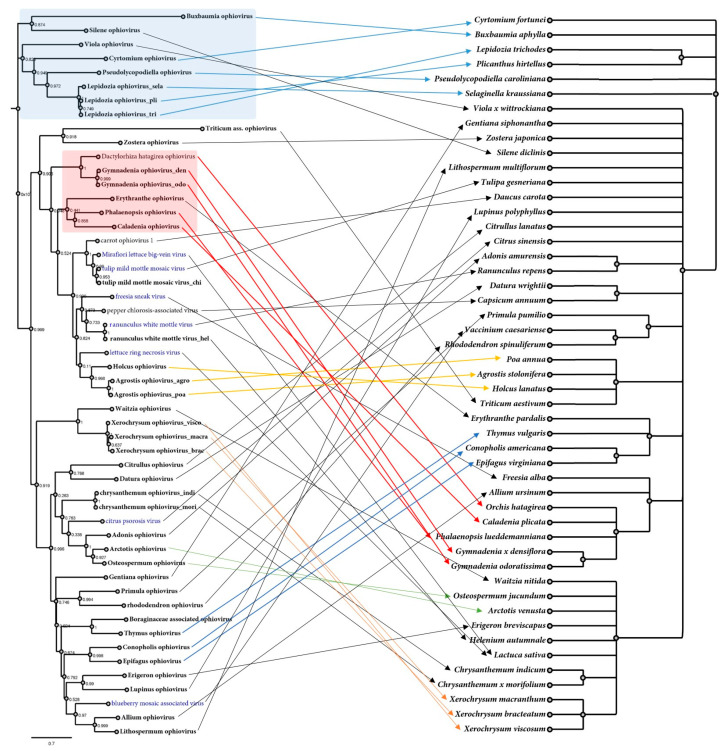
Tanglegram showing the phylogenetic relationships of the ophioviruses (left), which are linked with the associated plant host(s) shown on the right. Links of well-supported clades of viruses to taxonomically related plant species are indicated in colors. A maximum-likelihood phylogenetic tree of ophioviruses was constructed based on the CP protein. Plant host cladograms were generated in phyloT v.2 based on NCBI taxonomy. Viruses identified in the present study are shown in bold font. Two clusters mostly represented by viruses detected in basal plants such as mosses, liverworts and ferns and a second one of orchid-associated viruses are indicated by light blue and light red rectangles, respectively. Viruses corresponding to members of ICTV-recognized species are depicted in blue. The scale bar indicates the number of substitutions per site.

**Table 1 viruses-15-00840-t001:** Summary of novel ophioviruses identified from plant RNA-seq data available in the NCBI database. Acronyms of best hits are listed in Appendix A.

Plant Host	Taxa/ Family	Virus Name/ Abbreviation	Bioproject ID/ Data Citation	Segment/Virus Reads (Total/RPKM)	Length (nt)	Accession Number	Protein ID	Length (aa)	Highest-Scoring Virus Protein	Blastp E-Value	Blastp Query Coverage%/	Blastp Identity%
Amur adonis (*Adonis amurensis*)	dicot/ *Ranunculaceae*	Adonis ophiovirus/ AdoOV	PRJNA521968/ [17]	RNA1 (693/2.1) RNA2 (1032/14.6) RNA3 (2406/37.5)	7425 * 1595 1448	BK062646 BK062647 BK062648	RdRp MP CP	2411 * 467 450	CPsV-RdRp CPsV-MP CPsV-CP	0.0 3e−108 6e−110	93 94 100	46.79 40.44
Creeping bentgrass (*Agrostis stolonifera*)	monocot/ *Poaceae*	Agrostis ophiovirus_agro/ AgrOV_agro	PRJNA324407/ [18]	RNA1 (403/0.2) RNA2 (214/0.4) RNA3 (404/0.9) RNA4 (269/0.5)	7710 * 1863 1540 1907	BK062649 BK062650 BK062651 BK062652	RdRp 22kDa MP CP 37kDa	2294 * 174 499 453 322	MLBVV-RdRp RWMV-22kDa MLBVV-MP MLBVV-CP LNRV-37kDa	0.0 1e−21 1e−155 3e−138 4e−69	84 75 98 99 90	59.61 40.91 49 49 37.54
Annual bluegrass (*Poa annua*)	monocot/ *Poaceae*	Agrostis ophiovirus_poa/ AgrOV_poa	PRJNA265116/ [19]	RNA 1 (36/0.2) RNA2 (903/1.9) RNA3 (932/2.4) RNA4 (206/0.6)	824 * 1828 1525 1428	BK062653 BK062654 BK062655 BK062656	22kDa MP CP 37kDa	174 499 453 323	RWMV-22kDa MLBVV-MP MLBVV-CP LRNV-37kDa	7e−21 5e−155 1e−138 7e−69	77 98 99 90	40.96 49 49 37.59
Wild garlic (*Allium ursinum*)	monocot/ *Amaryllidaceae*	Allium ophiovirus/ AllOV	PRJNA542932/ [20]	RNA1 (15956/14.8) RNA2 (11305/42.3) RNA3 (16494/75.6)	7380 * 1832 1495	BK062657 BK062658 BK062659	RdRp MP CP	2338 478 454	BlMaV-RdRp BlMaV-MP BlMaV-CP	0.0 4e−111 5e−125	99 93 81	52.96 38 49.6
Silver actotis (*Arctotis venusta*)	dicot/ *Asteraceae*	Arctotis ophiovirus/ ActOP	PRJNA371565/ [21]	RNA1 (30281/99.0) RNA2 (18388/287.8) RNA3 (4558/84.8)	8319 1738 1462	BK062660 BK062661 BK062662	RdRp 22kDa MP CP	2406 222 486 446	CPsV-RdRp no hits CPsV-MP CPsV-CP	0.0 - 2e−142 1e−121	99 - 97 100	45.18 48 43.56
Borage (*Boranginaceae*)	dicot/ *Boranginaceae*	Boranginaceae associated ophiovirus/BaOV	PRJNA659133/ [22]	RNA2 (625/4.8) RNA3 (1684/14.2)	1737 1589	BK062663 BK062664	MP CP	486 471	BlMaV-MP BlMaV-CP	2e−110 2e−86	91 76	39.74 41.92
Bug moss (*Buxbaumia aphylla*)	Bryophyta/ *Buxbaumiaceae*	Buxbaumia ophiovirus/ BuxOV	PRJEB21674/ 1000 Plant (1KP) Transcriptomes Initiative	RNA3 (1296/27.8)	1590	BK062665	CP	485	MLBVV-CP	4e−18	53	25.82
Crab-lipped spider orchid (*Caladenia plicata*)	monocot/ *Orchidaceae*	Caladenia ophiovirus/ CalOV	PRJNA384875/ [23]	RNA1 (5541/15.3) RNA2 (1126/13.2) RNA3 (447/6.5)	7488 1760 1423	BK062666 BK062667 BK062668	RdRp 22kDa MP CP	2247 163 445 438	RWMV-RdRp RWMV-22kDa LRNV-MP RWMV-CP	0.0 7e−10 9e−100 4e−101	100 61 99 94	49.48 32.67 39.47 39.66
Indian chrysanthemum (*Chrysanthemum indicum*)	dicot/ *Asteraceae*	Chrysanthemum ophiovirus_indi/ChrOV_indi	PRJNA361213/ [24]	RNA1 (1318/2.8) RNA2 (1242/10.1) RNA3 (597/6.6)	8240 2143 1572	BK062669 BK062670 BK062671	RdRp 22kDa MP CP	2379 222 483 457	BlMaV-RdRp BlMaV-22kDa CPsV-MP BlMaV-CP	0.0 0.002 4e−117 3e−109	97 62 94 99	/47.45 30.54 42.61 42.19
Garden mum (*Chrysanthemum morifolium*)	dicot/ *Asteraceae*	Chrysanthemum ophiovirus_mori/ChrOV_mori	PRJNA315793/ [25]	RNA1 (12382/4.8) RNA2 (4371/6.5) RNA3 (1635/3.3)	8255 2164 1573	BK062672 BK062663 BK062674	RdRp 22kDa MP CP	2379 222 483 457	BlMaV-RdRp BlMaV-22kDa CPsV-MP BlMaV-CP	0.0 0.002 4e−117 3e−109	97 62 94 99	47.49 30.50 42.64 42.15
Watermelon (*Citrullus lanatus*)	dicot/ *Cucurbitaceae*	Citrullus ophiovirus/CitOV	PRJNA576654/ [26]	RNA1 (10212/21.4) RNA2 (32771/332.7) RNA3 (18763/219.4)	8510 1760 1528	BK062675 BK062676 BK062677	RdRp 22kDa MP CP	2418 245 483 464	CPsV-RdRp no hits CPsV-MP CPsV-CP	0.0 - 9e−127 8e−82	99 - 97 93	42.77 43.19 38.66
Bear corn (*Conopholis americana*)	dicot/ *Orobanchaceae*	Conopholis ophiovirus/ConOV	PRJEB21674/ 1000 Plant (1KP) Transcriptomes Initiative	RNA3 (1148/22.5)	1684	BK062678	CP	481	BlMaV-CP	4e−99	69	45.35
Holly fern (*Cyrtomium fortunei*)	*Polypodiophyta*/ *Dryopteridaceae*	Cyrtomium ophiovirus/CyrOV	PRJNA384992/ [27]	RNA1 (16605/59.5) RNA2 (18411/261.7) RNA3 (42710/660.2)	7548 1902 1749	BK062679 BK062680 BK062681	RdRp 22kDa MP CP	2357 105 409 500	BlMaV-RdRp no hits BlMaV-MP MLBVV-CP	0.0 - 4e−06 4e−50	97 - 67 67	36.83 22.92 33.90
Sacred datura (*Datura wrightii*)	Dicot/ *Solanaceae*	Datura ophiovirus/DatOV	PRJNA473174/ Sun, University of California, USA	RNA1 (3286/13.0) RNA2 (6830/122.1) RNA3 (18608/349.7)	8055 1788 1701	BK062682 BK062683 BK062684	RdRp 22kDa MP CP	2366 186 481 511	BlMaV-RdRp no hits BlMaV-MP CPsV-CP	0.0 - 1e−86 1e−88	97 - 95 69	50.09 35.43 39.50
Beech drops (*Epifagus virginiana*)	dicot/ *Orobanchaceae*	Epifagus ophiovirus/EpiOV	PRJEB21674/ 1000 Plant (1KP) Transcriptomes Initiative	RNA3 (250/7.8)	1371 *	BK062685	CP	328 *	BlMaV-CP	3e−59	88	42.81
Lifeflower (*Erigeron breviscapus*)	dicot/ *Asteraceae*	Erigeron ophiovirus/EriOV	PRJNA293262/ [28]	RNA3 (817/2.1)	1837	BK062686	CP	463	BlMaV-CP	2e−92	74	44.09
Pardus monkey-flower (*Erythranthe pardalis*)	dicot/ *Phrymaceae*	Erythranthe ophiovirus/EryOV	PRJNA508749/ [29]	RNA1 (611/2.6) RNA2 (539/11.0) RNA3 (3995/78.3)	7643 1587 1651	BK062687 BK062688 BK062689	RdRp 22kDa MP CP	2271 195 436 490	LRNV-RdRp BlMaV-22kDa LRNV-MP RWMV-CP	0.0 2e−11 1e−97 1e−100	99 58 90 99	51.73 30.43 39.33 36.68
Tube gentian (*Gentiana siphonantha*)	dicot/ *Gentianaceae*	Gentiana ophiovirus/ (GenOV)	PRJNA555883/ [30]	RNA1 (9973/23.4) RNA2 (1620/14.7) RNA3 (1561/20.0)	8043 2077 1473	BK062690 BK062691 BK062692	RdRp 22kDa MP CP	2254 190 516 450	BlMaV-RdRp BlMaV-22kDa BlMaV-MP BlMaV-CP	0.0 0.007 3e−47 2e−92	99 75 52 80	45.32 24.32 37.86 42.66
Marsh fragrant orchid (*Gymnadenia densiflora*)	monocot/ *Orchidaceae*	Gymnadenia ophiovirus_den/GymOV_den	PRJNA504609/ [31]	RNA3 (336/5.6)	1431	BK062693	CP	446	DhOV-CP	0.0	94	60.05
Short-spurred fragrant orchid (*Gymnadenia odorattissima*)	monocot/ *Orchidaceae*	Gymnadenia ophiovirus_odo/GymOV_odo	PRJNA504609/ [31]	RNA3 (172/3.0)	1339 *	BK062694	CP	425	DhOV-CP	0.0	87	58.25
Common velvetgrass (*Holcus lanatus*)	monocot/ *Poaceae*	Holcus ophiovirus/HolOV	PRJEB3994/ [32]	RNA 1 (2879/3.1) RNA2 (3959/18.5) RNA3 (3501/19.4) RNA4 (2272/13.1)	7627 * 1770 1495 1436	BK062695 BK062696 BK062697 BK062698	RdRp 22kDa MP CP 37kDa	2194 * 162 459 444 322	RWMV-RdRp MLBVV-22kDa MLBVV-MP RWMV-CP LRNV-37kDa	0.0 3e−24 2e−176 4e−148 1e−73	89 82 99 100 98	65.13 40 53.43 49.77 39.06
Hairy liverwort (*Lepidozia trichodes*)	*Marchantiophyta*/ *Lepidoziaceae*	Lepidozia ophiovirus_tri/LepOV_tri	PRJNA505755/ Fairylake Botanical Garden, China	RNA1 (38067/93.9) RNA2 (10558/106.3) RNA3 (28205/336.2)	7644 1872 1581	BK062699 BK062700 BK062701	RdRp 22kDa MP CP	2357 109 460 471	BlMaV-RdRp no hits BlMaV-MP MLBVV-CP	0.0 - 3e−19 6e−58	96 - 49 71	37.15 25.96 30.99
Basket liverwort (*Plicanthus hirtellus*)	*Marchantiophyta*/ *Anastrophyllaceae*	Lepidozia ophiovirus_pli/LepOV_pli	PRJNA505755/ Fairylake Botanical Garden, China	RNA1 (1358/3.8) RNA2 (128/1.8) RNA3 (1057/14.3)	7546 1497 1555	BK062702 BK062703 BK062704	RdRp 22kDa MP CP	2357 109 460 471	BlMaV-RdRp no hits BlMaV-MP MLBVV-CP	0.0 - 2e−19 8e−58	96 - 49 71	37.19 25.94 30.92
Krauss’ spike moss (*Selaginella kraussiana*)	Lycophyta/ *Selaginellaceae*	Lepidozia ophiovirus_sela/LepOV_sela	PRJNA351923/ [33]	RNA1 (556211 /499.8) RNA2 (75738/277.9) RNA3(288058/1251.5)	7644 1872 1581	BK062705 BK062706 BK062707	RdRp 22kDa MP CP	2357 109 460 471	BlMaV-RdRp no hits BlMaV-MP MLBVV-CP	0.0 - 4e−19 5e−58	96 - 49 71	37.11 25.99 30.95
Manyflowered gromwell (*Lithospermum multiflorum*)	dicot/ *Boraginaceae*	Lithospermum ophiovirus/LitOV	PRJNA353131/ [34]	RNA2 (449/6.1) RNA3 (2370/28.6)	1498 * 1693	BK062708 BK062709	MP CP	470* 460	BlMaV-MP MLBVV-CP	7e−118 1e−51	92 64	42.61 32.70
Garden lupin (*Lupinus polyphyllus*)	dicot/ *Fabaceae*	Lupinus ophiovirus/LupOV	PRJEB8056/ [35]	RNA3 (1631/45.8)	1838	BK062710	CP	448	BlMaV-CP	1e−95	73	44.31
Trailing pink daisy (*Osteospermum jucundum*)	dicot/ *Asteraceae*	Osteospermum ophiovirus/OstOV	PRJNA371565/ [21]	RNA1 (11077/35.9) RNA2 (11158/181.4) RNA3 (12821/234.5)	8521 1701 1512	BK062711 BK062712 BK062713	RdRp 22kDa MP CP	2407 204 482 449	CPsV-RdRp no hits CPsV-MP CPsV-CP	0.0 - 2e−143 2e−120	100 - 100 100	46.10 47.89 43.65
Moth orchid (*Phalaenopsis lueddemanniana*)	monocot/ *Orchidaceae*	Phalaenopsis ophiovirus/PhaOV	PRJNA345261/ [36]	RNA2 (709/8.6) RNA3 (1173/16.3) RNA4 (388/6.8)	1867 1630 1296	BK062714 BK062715 BK062716	MP CP 37kDa	489 431 360	LRNV-MP MLBVV-CP LRNV-37kDa	1e−92 7e−120 1e−14	95 100 47	37.58 44.87 31.76
Clammy primrose (*Primula pumilio*)	dicot/ *Primulaceae*	Primula ophiovirus/PriOV	PRJNA544345/ Hao, D., Chengdu, China	RNA2 (6291/82.6) RNA3 (8866/115.9)	1565 1572	BK062717 BK062718	MP CP	450 455	LRNV-MP CPsV-CP	2e−56 1e−86	91 93	30.64 37.15
Slender bog club-moss (*Pseudolycopodiella caroliniana*)	Lycophyta/ *Lycopodiaceae*	Pseudolycopodiella ophiovirus/PseOV	PRJEB4921/ 1000 Plant (1KP) Transcriptomes Initiative	RNA2 (695/20.5) RNA3 (1402/47.4)	1829 1594	BK062719 BK062720	MP CP	464 466	MLBVV-MP MLBVV-CP	1e−22 2e−55	57 71	28.37 31.86
Firecracker rhododendron (*Rhododendron spinuliferum*)	dicot/ *Ericaceae*	Rhododendron ophiovirus/RhoOV	PRJNA530078/ Xue Zhang, Yunnan University, China	RNA2 (1008/11.5) RNA3 (590/9.0)	1867 1406	BK062725 BK062726	MP CP	452 441	BlMaV-MP BlMaV-CP	3e−49 6e−78	95 95	31.25 37.07
Diclinis campion (*Silene diclinis*)	dicot/ *Caryophyllaceae*	Silene ophiovirus/ SilOV	PRJEB39526/ [37]	RNA1 (382/2.8) RNA2 (438/12.7) RNA3 (250/7.2)	6036 * 1511 1532	BK062727 BK062728 BK062729	RdRp MP CP	1993 * 426 446	BlMaV-RdRp LRNV-MP TMMMV-CP	0.0 2e−29 2e−53	98 69 80	40.51 29.30 35.12
Thyme (*Thymus vulgaris*)	dicot/ *Lamiaceae*	Thymus ophiovirus/ ThyOV	PRJNA417241/ [38]	RNA2 (1885/24.2) RNA3 (38453/495.5)	1598 1589	BK062730 BK062731	MP CP	480 477	BlMaV-MP BlMaV-CP	3e−114 1e−91	95 71	40.47 42.98
Wheat (*Triticum aestivum*)	monocot/ *Poaceae*	Triticum associated ophiovirus/TriaOV	PRJNA432496/ [39]	RNA1 (17636/41.4) RNA3 (476/5.0)	5377 * 1192 *	BK062733 BK062734	RdRp CP	1792 * 377 *	RWMV-RdRp DhOV-CP	0.0 8e−15	95 57	34.62 30.77
Pansies (*Viola x wittrockiana*)	dicot/ Violaceae	Viola ophiovirus/VioOV	PRJNA552204/ [40]	RNA1 (2761/5.5) RNA2 (188/1.8) RNA3 (126/1.2)	7671 1570 1576	BK062735 BK062736 BK062737	RdRp 22kDa MP CP	2308 173 435 492	MLBVV-RdRp no hits BlMaV-MP MLBVV-CP	0.0 - 4e−16 4e−52	94 - 54 69	37.76 27.17 33.62
Golden waitzia (*Waitzia nitida*)	dicot/ *Asteraceae*	Waitzia ophiovirus /(WaiOV)	PRJNA371565/ [21]	RNA2 (219/1.8) RNA3 (208/1.8)	1570 1486	BK062738 BK062739	MP CP	453 460	BlMaV-MP CPsV-CP	5e−35 7e−63	83 97	29.84 31.32
Strawflower (*Xerochrysum bracteatum*)	dicot/ *Asteraceae*	Xerochrysum ophiovirus_brac/ XerOV_brac_	PRJNA371565/ [21]	RNA1 (15362/46.6) RNA2 (7601/112.2) RNA3 (11398/169.0)	7681 1577 1570	BK062740 BK062741 BK062742	RdRp 22kDa MP CP	2266 199 444 461	BlMaV-RdRp no hits LRNV-MP CPsV-CP	0.0 - 4e−39 7e−56	99 - 90 94	40.75 28.76 30.16
White strawflower (*Xerochrysum macranthum*)	dicot/ *Asteraceae*	Xerochrysum ophiovirus_macra/ XerOV_macra	PRJNA371565/ [21]	RNA1 (3999/13.0) RNA2 (5003/76.2) RNA3 (2662/44.1)	7692 1646 1513	BK062743 BK062744 BK062745	RdRP 22kDa MP CP	2264 199 444 459	BlMaV-RdRp no hits LRNV-MP CPsV-CP	0.0 - 4e−40 7e−57	99 - 90 95	40.75 29.82 30.11
Sticky everlasting (*Xerochrysum viscosum*)	dicot/ *Asteraceae*	Xerochrysum ophiovirus_visco/ XerOV_visco	PRJNA371565/ [21]	RNA1 (4099/13.7) RNA2 (346/5.5) RNA3 (304/5.1)	7591 1577 1522	BK062746 BK062747 BK062748	RdRp 22kDa MP CP	2266 199 441 459	BlMaV-RdRp no hits LRNV-MP CPsV-CP	0.0 - 2e−41 3e−58	0.0 - 91 96	41.45 30.05 30.44
Dwarf eelgrass (*Zostera japonica*)	monocot/ *Zosteraceae*	Zostera ophiovirus/ ZosOV	PRJNA419030/ [41]	RNA1 (42459/165.7) RNA3 (15392/284.8)	7748 1634	BK062749 BK062750	RdRp 22kDa CP	2281 216 452	LRNV-RdRp no hits RWMV-CP	0.0 - 4e−52	93 - 99	41.84 30.31

* Partial sequence (predicted coding region is incomplete/truncated).

**Table 2 viruses-15-00840-t002:** Novel viruses: virus name and tentative species names within genus *Ophiovirus*.

Virus Name/Abbreviation	Species Name
Adonis ophiovirus/AdoOV	*Ophiovirus adonidis*
Agrostis ophiovirus_agro/AgrOV_agro	*Ophiovirus agrostis*
Agrostis ophiovirus_poa/AgrOV_poa	*Ophiovirus agrostis*
Allium ophiovirus/AllOV	*Ophiovirus alli*
Arctotis ophiovirus/ActOP	*Ophiovirus arctotis*
Boranginaceae associated ophiovirus/BaOV	*Ophiovirus* *boranginaceae*
Buxbaumia ophiovirus/BuxOV	*Ophiovirus buxbaumiae*
Caladenia ophiovirus/CalOV	*Ophiovirus caladeniae*
chrysanthemum ophiovirus_indi/ChrOV_indi	*Ophiovirus chrysanthemi*
chrysanthemum ophiovirus_mori/ChrOV_mori	*Ophiovirus chrysanthemi*
Citrullus ophiovirus/CitOV	*Ophiovirus citrullus*
Conopholis ophiovirus/ConOV	*Ophiovirus conopholis*
Cyrtomium ophiovirus/CyrOV	*Ophiovirus cyrtomii*
Datura ophiovirus/DatOV	*Ophiovirus daturi*
Epifagus ophiovirus/EpiOV	*Ophiovirus epifagus*
Erigeron ophiovirus/EriOV	*Ophiovirus erigeron*
Erythranthe ophiovirus/EryOV	*Ophiovirus erythranthis*
Gentiana ophiovirus/ (GenOV)	*Ophiovirus gentianae*
Gymnadenia ophiovirus_den/GymOV_den	*Ophiovirus gymnadeniae*
Gymnadenia ophiovirus_odo/GymOV_odo	*Ophiovirus gymnadeniae*
Holcus ophiovirus/HolOV	*Ophiovirus holci*
Lepidozia ophiovirus_tri/LepOV_tri	*Ophiovirus lepidoziae*
Lepidozia ophiovirus_pli/LepOV_pli	*Ophiovirus lepidoziae*
Lepidozia ophiovirus_sela/LepOV_sela	*Ophiovirus lepidoziae*
Lithospermum ophiovirus/LitOV	*Ophiovirus lithospermi*
Lupinus ophiovirus/LupOV	*Ophiovirus lupini*
Osteospermum ophiovirus/OstOV	*Ophiovirus osteospermi*
Phalaenopsis ophiovirus/PhaOV	*Ophiovirus phalaenopsis*
Primula ophiovirus/PriOV	*Ophiovirus primuli*
Pseudolycopodiella ophiovirus/PseOV	*Ophiovirus pseudolycopodiellae*
rhododendron ophiovirus/RhoOV	*Ophiovirus rhododendri*
Silene ophiovirus/SilOV	*Ophiovirus sileni*
Thymus ophiovirus/ ThyOV	*Ophiovirus thymi*
Triticum associated ophiovirus/TriaOV	*Ophiovirus tritici*
Viola ophiovirus/VioOV	*Ophiovirus violae*
Waitzia ophiovirus /(WaiOV)	*Ophiovirus waitziae*
Xerochrysum ophiovirus_brac/ XerOV_brac_	*Ophiovirus xerochrysi*
Xerochrysum ophiovirus_macra/ XerOV_macra	*Ophiovirus xerochrysi*
Xerochrysum ophiovirus_visco/ XerOV_visco	*Ophiovirus xerochrysi*
Zostera ophiovirus/ZosOV	*Ophiovirus zosterae*

## Data Availability

Nucleotide sequence data reported are available in the Third Party Annotation Section of the DDBJ/ENA/GenBank databases under the accession numbers TPA: BK062646-BK062750 and can be found as in the Appendix A of this article.

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
