# Peer review of "Expanding the Repertoire of the Plant-Infecting Ophioviruses through Metatranscriptomics Data"

_viruses, 2023, doi:10.3390/v15040840_

Round 1

Author Response

The manuscript of Debat et al. describes the identification and sequence characterization of ophioviruses through in silico analysis of metatranscriptomic data. The knowledge on ophioviruses is still limited, from this point, I consider the work in this manuscript as interesting and novel.

We thank reviewer #1 for taking the time to thoroughly assess our MS and provide suggestions which improved the MS.

I would suggest focusing on the data that are complete and convincing, instead of putting everything in this manuscript. For instance, the authors can focus on the 12 full length viral genomes sequences or the complete segments/genes that they obtained in this study. In addition, authors need to represent their data and rewrite some sections to efficiently deliver the information.

Thank you. We have re organized and re drafted the manuscript, we have separated results and discussions and included subsections to improve the clarity and organization of the MS. We have focused in the new discoveries and avoided redundancies. We believe the MS in its present form is easier to follow.

There are a lot of numbering used here which I found were rather confusing, such as in the following part: “In this study, 12 tentatively full-length viral genome sequences were obtained through the identification, assembly, and curation of raw NCBI-SRA reads. Additionally, five of the putative viruses had all their RNA segments annotated, while 16 had some missing, mostly derived from technical difficulties to assemble segments which are relatively at low RNA levels during infection such as RNA 1.”

We have completely re drafted this section (and others) to avoid confusions.

Although the authors provide nucleotide sequence information in the supplementary document, the sequences should have been deposited in the GenBank by the time of manuscript submission. This is because I found that the accession numbers were not found in the GenBank.

Sequence were deposited in GenBank a couple of months ago (before submitting this MS). We have obtained and indicated in the data availability statement the corresponding accession numbers. In addition, we have already requested to GenBank to release the sequences, which will be promptly online.

Some of the viruses contain “N” strings in their sequence, indicating that the identify of these nucleotides cannot be determined. But these sequences were still included for analysis. For example, the RNA1 of adonis ophiovirus was used for phylogenetic analysis of the L protein although it only has partial sequence with ambiguities.

Genetic identities assessments and main phylogenetic and evolutionary insights were generated with the CP protein which were complete or nearly complete. In the case of L, we have added some sequences that include ambiguities which are masked by the analysis pipeline but only to show that it mimics the results obtained using CP. We have added a sentence for clarification.  

Are there any minimum cut off value that the authors used to determine whether the sequence belongs to an ophiovirus?

We have included the available molecular criterion for species demarcation criteria for ophioviruses, and how our detections would accommodate to this threshold. In addition our main cut off indicator used for our detections was that the best hit of the predicted product of each segment should be a protein of a known ophiovirus (and not a virus of the unclassified aspi-like group which are most probably linked to fungi). That being said, taxonomy here is taken cautiously and will be a task for the corresponding ICTV subcommittee. We have added a sub section to introduce these matters.

Motifs and domains of the respective predicted proteins should be clearly indicated in the sequence in the form of figure etc.

As suggested, we have redrafted Figure 1 to include predicted proteins and domains.

The authors mentioned that they used reference-based mapping to longer contigs, but data such as the percentage of mapped reads, average read coverage etc were not shown.

A column in Table 1 was added to give a measure of robustness of predictions. Basically total virus reads used for the assembly and RPKM of the corresponding library.

The significant clades in the phylogenetic tree which correspond to their respective hosts such as orchid, fern etc. should be highlighted.

We have redrafted Figure 3 to indicate significant clades of the tree.

The usage of proteins in figure and in main text. (E.g., Supplementary figure 3) should also be standardized.

Done.

Reviewer 2 Report

By analyzing public data, this paper identifies some novel ophioviruses that may be new virus species in the genus. Phylogenetic analyses showed a novel clade of mosses, liverworts and fern ophioviruses, characterized by long branches suggesting still plenty unsampled hidden diversity within the genus. This study represents a significant expansion of genomics of ophioviruses. Only a concern, as stated by the author in the discussion, is that the incapacity to return to the original biological material to repeat and check the assembled viral genome sequences.  In addition, please check "Koonin et al., 2021" in the introduction , not be found in the references.  the "results and discussion " is too long, which may be better to describe it in separate paragraphs.

Author Response

By analyzing public data, this paper identifies some novel ophioviruses that may be new virus species in the genus. Phylogenetic analyses showed a novel clade of mosses, liverworts and fern ophioviruses, characterized by long branches suggesting still plenty unsampled hidden diversity within the genus. This study represents a significant expansion of genomics of ophioviruses.

We thank reviewer #2 for taking the time to thoroughly assess our MS and provide suggestions which improved the MS.

Please check "Koonin et al., 2021" in the introduction, not be found in the references. 

We added this reference.

The "results and discussion " is too long, which may be better to describe it in separate paragraphs.

Thank you. We have re organized and re drafted the manuscript, we have separated results and discussions and included subsections to improve the clarity and organization of the MS.

Reviewer 3 Report

The manuscript titled "Expanding the Repertoire of the Plant-Infecting Ophioviruses" provides a refreshing reminder of the limitations of virus taxonomy. As mentioned in this manuscript, the Genus ophiovirus currently only catalogues seven viruses. This manuscript has the potential to expand the repertoire of ophioviruses several times over. Although data-based exploration cannot directly confirm the presence of a plant ophiovirus, the predictions it provides offer substantial clues. Moreover, these methods could be employed to provide clues for discovering new viruses in other families. Finally, the manuscript was well-written, concise, and well-organized, one of the best manuscripts that I have ever reviewed, with only a few typos that need correction, such as "(some?)".

Author Response

The manuscript titled "Expanding the Repertoire of the Plant-Infecting Ophioviruses" provides a refreshing reminder of the limitations of virus taxonomy. As mentioned in this manuscript, the Genus ophiovirus currently only catalogues seven viruses. This manuscript has the potential to expand the repertoire of ophioviruses several times over. Although data-based exploration cannot directly confirm the presence of a plant ophiovirus, the predictions it provides offer substantial clues. Moreover, these methods could be employed to provide clues for discovering new viruses in other families. Finally, the manuscript was well-written, concise, and well-organized, one of the best manuscripts that I have ever reviewed

We thank reviewer #3 for taking the time to thoroughly assess our MS and provide suggestions which improved the MS.

Only a few typos that need correction, such as "(some?)"

Thank you. We have corrected many typos including the one indicated by reviewer #3.